# Evolutionary conservation and post-translational control of S-adenosyl-L-homocysteine hydrolase in land plants

**Sara Alegre**[1], **Jesús Pascual**[1], **Andrea Trotta**[1,2], **Martina Angeleri**[1], **Moona Rahikainen**[1], **Mikael Brosche**[3], **Barbara Moffatt**[4], **Saijaliisa Kangasjärvi**[1]*

**1** Department of Biochemistry, Molecular Plant Biology, University of Turku, Turku, Finland, **2** Institute of Biosciences and Bioresources, National Research Council of Italy, Sesto Fiorentino, Firenze, Italy, **3** Organismal and Evolutionary Biology Research Program, Faculty of Biological and Environmental Sciences, Viikki Plant Science Centre, University of Helsinki, Helsinki, Finland, **4** Department of Biology, University of Waterloo, Waterloo, Ontario, Canada

* saijaliisa.kangasjarvi@utu.fi

**Data Availability Statement:** All relevant data are within the manuscript and its Supporting Information files.

## Abstract

Trans-methylation reactions are intrinsic to cellular metabolism in all living organisms. In land plants, a range of substrate-specific methyltransferases catalyze the methylation of DNA, RNA, proteins, cell wall components and numerous species-specific metabolites, thereby providing means for growth and acclimation in various terrestrial habitats. Trans-methylation reactions consume vast amounts of S-adenosyl-L-methionine (SAM) as a methyl donor in several cellular compartments. The inhibitory reaction by-product, S-adeno-syl-L-homocysteine (SAH), is continuously removed by SAH hydrolase (SAHH), which essentially maintains trans-methylation reactions in all living cells. Here we report on the evolutionary conservation and post-translational control of SAHH in land plants. We provide evidence suggesting that SAHH forms oligomeric protein complexes in phylogenetically divergent land plants and that the predominant protein complex is composed by a tetramer of the enzyme. Analysis of light-stress-induced adjustments of SAHH in *Arabidopsis thaliana* and *Physcomitrella patens* further suggests that regulatory actions may take place on the levels of protein complex formation and phosphorylation of this metabolically central enzyme. Collectively, these data suggest that plant adaptation to terrestrial environments involved evolution of regulatory mechanisms that adjust the trans-methylation machinery in response to environmental cues.

## Introduction

Land plants have evolved sophisticated biochemical machineries that support cell metabolism, growth and acclimation in various terrestrial habitats. One of the most common biochemical modifications occurring on biological molecules is methylation, which is typical for DNA, RNA, proteins, and a vast range of metabolites. Trans-methylation reactions are therefore important in a relevant number of metabolic and regulatory interactions, which determine

**Funding:** This work was financially supported by Academy of Finland (www.aka.fi) project 307719 to SK, 325122 to the salary of JP, and the Academy of Finland Center of Excellence in Primary Producers 2014-2019 (307335). SA and MR received salary from the University of Turku Doctoral Programme in Molecular Life Sciences (https://www.utu.fi/en/research/utugs/doctoral-programme-in-molecular-life-sciences). MR also received salary from the Turku University Foundation (https://www.yliopistosaatio.fi/en/) and the Finnish Cultural Foundation Varsinais-Suomi Regional Fund (https://skr.fi/en/regional-funds/varsinais-suomi-regional-fund). MB was funded by the University of Helsinki (www.helsinki.fi). The funders had no role in study design, data collection and analysis, decision to publish, or preparation of the manuscript.

**Competing interests:** The authors have declared that no competing interests exist.

physiological processes during the entire life cycle of plants. Trans-methylation reactions are carried out by methyl transferases (MTs), which can be classified into O-MTs, N-MTs, C-MTs and S-MTs based on the atom that hosts the methyl moiety [1,2]. All these enzymes require S-adenosyl-L-methionine (SAM) as a methyl donor [3]. Among MTs, O-MTs form a large group of substrate-specific enzymes capable of methylating RNA, proteins, pectin, mono-lignols as well as various small molecules in different cellular compartments [2].

The availability of SAM is a prerequisite for methylation, while the methylation reaction by-product, S-adenosyl-L-homocysteine (SAH), which competes for the same binding site on the MT, is a potent inhibitor of MT activity and must therefore be efficiently removed [4]. To ensure the maintenance of SAM-dependent trans-methylation capacity, SAH is rapidly hydro-lysed by S-adenosyl-L-homocysteine hydrolase (SAHH, EC 3.3.1.1) in a reaction that yields L-homocysteine (HCY) and adenosine (ADO) [5]. Subsequently, methionine is regenerated from HCY by cobalamin-independent methionine synthase (CIMS, EC 2.1.1.14) using methyl-tetrahydrofolate as a methyl donor. Finally, methionine is converted to SAM in an ATP-dependent reaction driven by SAM synthase, also known as methionine adenosyltransferase (SAMS/MAT, EC 2.5.1.6). This set of reactions are collectively termed as the Activated Methyl Cycle (AMC).

SAHH is the only known eukaryotic enzyme capable of hydrolysing SAH, and therefore a key player in the maintenance of cellular transmethylation potential. The *Arabidopsis thaliana* genome encodes two SAHH isoforms, SAHH1 (AT4G13940) and SAHH2 (AT3G23810) and particularly SAHH1 is indispensable for physiological functions at different developmental stages [6–8]. Null mutation of SAHH1 has been reported embryo lethal in *A. thaliana* and severe symptoms caused by SAHH deficiency have been reported in human [6,9,10] whereas there were no morphological abnormalities in homozygous *A. thaliana sahh2* T-DNA insertion mutants [6]. Mutants suffering from impaired SAHH1 function, including the knockdown *sahh1* and *homology-induced gene silencing 1* (*hog1*), were viable but showed delayed germination, slow growth and short primary roots [6,11]. Evidently, SAHH is crucial in ensuring accurate metabolic and regulatory reactions in the cell. Even though a number of post-translational modifications (PTMs) has been detected on *A. thaliana* SAHH [2], the exact regulatory mechanisms remain poorly understood.

At the amino acid sequence level, SAHH is one of the most highly conserved enzymes across the kingdoms of life [12]. Crystallography and structural studies from phylogenetically distant species have reported SAHH to form dimers, tetramers and hexamers in plant, mammalian and bacterial species [13–16]. The high-resolution crystal structure of *Lupinus luteus* SAHH1 suggested that in higher plants the enzyme would form functional dimers with a calculated molecular mass of 110 kDa [14,15]. However, in *A. thaliana* leaves SAHH1 and SAHH2 were predominantly detected in an oligomeric protein complex called SAHH complex 4, which has an approximate molecular weight of 200 kDa [17]. The subunit composition of this abundant oligomeric SAHH complex has not been resolved and potential controversy behind these observations therefore remains unclear. It should be noted, however, that the pioneering work on the *L. luteus* SAHH1 was established by *in vitro* studies using gel filtration and crystallography approaches, and the structural studies were conducted with a recombinant, not post-translationally modified enzyme [14,15].

Here we report on the evolutionary conservation and biochemical characteristics of SAHH in land plants. We find that a predominant oligomeric SAHH complex with similar apparent molecular weight as *A. thaliana* SAHH complex 4 can be detected in phylogenetically divergent land plants, and provide evidence suggesting that in *A. thaliana* the protein complex is a tetrameric form of the enzyme. Regulatory adjustments observed on SAHH in high-light-exposed *A. thaliana* and *Physcomitrella patens* further suggest that both angiosperms and

bryophytes respond to light-induced stress by regulatory adjustments in this metabolically central enzyme.

## Materials and methods

### Plant material

*Arabidopsis thaliana* wild type accession Columbia-0, a transgenic *A. thaliana* line stably expressing *SAHH1p::EGFP-SAHH1* [18], *Brassica oleracea* convar. *acephala* varieties Half Tall and Black Magic (kales) and *Lupinus luteus* were grown in peat:vermiculite (2:1) and 50% relative humidity at 8-hour light period under 130 µmol photons $m^{-2}$ $s^{-1}$ and 22˚C. Samples were collected after 4 weeks of growth. *Spinacia oleracea* (spinach) and *Brassica oleracea* convar. *italica* (broccoli) were purchased from the local supermarket. *Physcomitrella patens* was grown for 13 days on agar plates in minimum media [19] in a 16-hour photoperiod under 45 µmol photons $m^{-2}$ $sec^{-1}$ at 24˚C. For high light stress experiments, *A. thaliana* was grown for 16 days in a 12-hour light period under 130 µmol photons $m^{-2}$ $s^{-1}$ and thereafter shifted to 800 µmol photons $m^{-2}$ $s^{-1}$ at 26˚C at a 12-hour light period for 2 days. *P. patens* was grown as described above and shifter to 500 µmol photons $m^{-2}$ $s^{-1}$ in a 16-hour light period for 2 days.

### Analysis of publicly available transcript profiles

O-methyltransferases were selected from the UniProt database (https://www.uniprot.org/) (September 2019) using the following search criteria: "O-methyltransferases" + "*Arabidopsis thaliana*" + "reviewed". This list was supplemented with the Activated Methyl Cycle enzymes according to Rahikainen et al. [2]. Together, the selected O-MTs and AMC enzymes formed a total of 40 genes, which were used as the input in GENEVESTIGATOR [20]. This input was assigned to 39 genes since AT5G17920 and AT3G03780, encoding CIMS1 and CIMS2, respectively could not be distinguished because they share the same probe in Affymetrix Arabidopsis ATH1 microarray. The database search was limited to "Only Columbia-0 Wild Type from Affymetrix Arabidopsis ATH1 genome array". The "perturbations" tool from GENEVESTIGATOR was used to determine in which experimental conditions the selected genes were differentially expressed. Experiments in which at least 60% (24 out of 39) of the input genes were differentially expressed (p-value <0.05) were selected for hierarchical clustering. The perturbations were hierarchically clustered with R package pheatmap (v1.0.12) [21] using Ward´s method and Euclidean distance.

### Confocal microscopy

Fluorescence from EGFP was imaged with a confocal laser scanning microscope Zeiss LSM780 with either C-Apochromat 40x/1.20 W Korr M27 or Plan-Apochromat 20x/0.8 objective. EGFP was excited at 488 nm and detected at 493 to 598 nm wave length and chlorophyll fluorescence was excited at 633 nm and detected at 647 to 721 nm wave length. Hectian strands were visualized by plasmolysis with 1 M NaCl and imaged after 6 minutes incubation. Images were created with Zeiss Zen 2.1 software version 11.0.0.190.

### Isolation of protein extracts and biochemical analysis of SAHH

Leaves of *A. thaliana*, *B. oleracea* convar. *acephala* (Half Tall and Black Magic), *B. oleracea* convar. *italica*, *L. luteus* and *S. oleracea*, and aerial parts of *P. patens* were ground in liquid nitrogen and mixed with extraction buffer [10 mM HEPES-KOH pH 7.5, 10 mM $MgCl_2$, supplemented with protease (Pierce EDTA-free Minitablets; Thermo Fisher Scientific) and

phosphatase (PhosSTOP; Roche) inhibitors]. The samples were centrifuged at 18,000 g for 15 minutes and the soluble fractions were taken for further analysis.

For biochemical analysis of *A. thaliana* SAHH complex 4, soluble protein fractions were treated with 0.25%, 1% sodium dodecyl sulfate (SDS; CAS Number 151-21-3) and/or 10 mM dithiothreitol (DTT; CAS Number 3483-12-3) in a total volume of 20 μL for 60 minutes as indicated in the figure legends. To assess protein complex formation, soluble protein fractions corresponding to 5 μg of protein were separated on Clear Native (CN) PAGE with a 7.5–12% gradient of acrylamide as in [17]. 2D-CN PAGE of *A. thaliana* and spinach leaf soluble proteins was performed with 90 μg of protein as in [22]. Protein spots on the 2D map were recognized based on their shape, position and intensity when compared to the 2D protein maps of *A. thaliana* soluble protein extracts reported previously [17,23].

For mass spectrometry analysis of *A. thaliana* and spinach SAHH-containing spots, excised spots were reduced, alkylated and digested with trypsin as in Rahikainen at al., 2017 [17]. Digested samples were analyzed in a nanoflow HPLC system (EasyNanoLC1000, Thermo Fisher Scientific) equipped with a 20 x 0.1 mm i.d. pre-column combined with a 150 mm x 75 μm i.d. analytical column, both packed with 5 μm Reprosil C18-bonded silica (Dr Maisch GmbH) and injection to an electrospray ionization (ESI) source coupled to a Q-Exactive (Thermo Fisher Scientific) mass spectrometer. Peptides were separated in a three-step 20-minute gradient: from 3% to 43% solvent B in 10 minutes, followed by an increase to 100% in 5 minutes, and 5 minutes of 100% solvent B. Samples were analyzed in Data Dependent Acquisition (DDA) mode. The top 10 most intense precursors in each scan (*m/z* 300–2000) were selected for higher-energy collisional dissociation (HCD) fragmentation using an exclusion window of 10 seconds. Protein identification was performed in Proteome Discoverer 2.2 using Mascot v. 2.4 and against the non-redundant *A. thaliana* proteome (TAIR10) appended with a collection of the most common contaminants in the case of *A. thaliana* and against "Viridiplantae" UniProtKB database in the case of spinach. Monoisotopic mass, a maximum of two missed cleavages, 10 ppm precursor mass tolerance, 0.02 Da fragment mass tolerance and charge $\geq 2 +$ were the settings used for the searches. Methionine oxidation, N-term acetylation and serine, threonine and tyrosine phosphorylation were allowed as dynamic modifications and cysteine carbamidomethylation as static. PhosphoRS filter and Decoy Database Search using 0.01% (strict) and 0.05% (relaxed) false discovery rate (FDR) confidence thresholds were used to validate the confidence of the identifications.

Isoelectric focusing (IEF) was performed as in [24]. Phos-tag gel electrophoresis (WAKO) with 7.5% (w/v) acrylamide in the separation gel was performed according to manufacturer's instructions (www.wako-chem.co.jp/english/labchem). SAHH was detected by immunoblotting with anti-SAHH antibody [18] or by using SYPRO as a protein stain as described in [22]. ADENOSINE KINASE (ADK) was detected by anti-ADK antibody as in [22]. The experiments were repeated at least three times and representative images are shown. Immunoblot intensities were acquired using the Fiji software [25]. T-test was applied to the numerical values in R Studio environment [26].

## Amino acid alignment and construction of phylogeny tree

SAHH amino acid sequences from *Arabidopsis thaliana* (accession AT4G13940; The Arabidopsis Information Resource, www.arabidopsis.org), *Lupinus luteus* (accession Q9SP37; https://www.uniprot.org), *Brassica oleracea* convar. *capitata*, (accession Bol033424; https://phytozome.jgi.doe.gov/pz/portal.html), *Spinacia oleracea* (accession A0A0K9RFV6; https://www.uniprot.org) and *Physcomitrella patens* (accession Pp3c19_13810V3.1;

https://phytozome-next.jgi.doe.gov/) were aligned using ClustalW Multiple Alignment [27] in BioEdit Version 7.2.5. *Brassica oleracea* convar. *acephala* amino acid sequence was unavailable, thus *Brassica oleracea* convar. *capitata* was used as its closest sequence available. Identities and similarities were reckoned in BioEdit by pairwise comparison with BLOSUM62 matrix. Phylogenetic tree was built with the same amino acid sequences plus human SAHH1 (accession P23526; https://www.uniprot.org) using Neighbor-Joining method [28] in MEGA7.

## Results

### Dynamics of O-MT mRNA abundance in *A. thaliana*

The accuracy and biochemical specificity of trans-methylation reactions stem from a high number of substrate-specific MTs, whose expression patterns can be highly responsive to both endogenous and exogenous signals. Here we focused on AMC enzymes and the O-MTs, which are well known for their functions in the methylation of small metabolites that accumulate upon environmental perturbations. To illustrate the dynamism of O-MT transcript abundance in *A. thaliana*, we performed an exploratory analysis using the "Perturbations" tool in GENEVESTIGATOR. Reviewed O-MTs from *A. thaliana* gene list were obtained from UniProt (www.uniprot.org; September 2019) and this list was supplemented with enzymes of the AMC (S1 Table). Perturbations in which at least 60% (24 out of 39) of the genes for the AMC enzymes and selected MTs with a p-value <0.05 were considered as differentially expressed and were selected to build the cluster heatmap.

As shown in Fig 1, hierarchical clustering of the gene expression data suggested dynamic adjustments in O-MT transcript abundance in response to a number of perturbations, which formed seven clusters. Clusters 6 and 7, branched out from the rest of clusters and included processes related to hormonal signalling and seed germination, respectively, while clusters 1 to 5 were comprised of perturbation categories related to light conditions, biotic, abiotic and chemical stress, and other physiological processes. These findings supported the view that O-MTs are highly regulated at the level of mRNA abundance and contribute to a multitude of metabolic processes in different compartments of plant cells.

### Sub-cellular localization of SAHH1 in *A. thaliana* leaves

Next we assessed the sub-cellular localization of *A. thaliana* SAHH1. Confocal microscopy imaging of leaves of four-week-old *A. thaliana* plants stably expressing an EGFP-SAHH1 fusion protein under the native SAHH1 promoter (*SAHH1p::EGFP-SAHH1*) [18], revealed that SAHH localized to multiple sub-cellular compartments (Fig 2). However, SAHH1 was not uniformly localized within the cells, but rather highly organized to various cellular structures. SAHH1 was found dynamically associated with cytoplasmic strands, along the plasma membrane, in punctate structures, and around chloroplasts (Fig 2, S1 Video). In line with a previous report [18], strong fluorescence arising from EGFP-SAHH1 was also detected in the nucleus (Fig 2E). The nucleolus, however, was completely devoid of SAHH1 (Fig 2E). Imaging of wild type leaves with the confocal microscopy settings used to detect EGFP did not reveal signals arising from potential autofluorescent compounds (S1 Fig, S2 Video, S2 Table). Immunoblot analysis of protein extracts separated on Clear Native (CN) gels revealed the presence of EGFP-SAHH1 in oligomeric protein complexes similar to those observed in wild type, and especially the abundant SAHH complex 4 (Fig 2G) [17]. Moreover, immunoblot analysis of SDS-gels confirmed the presence of EGFP-SAHH1 in the leaf extracts, whereas free EGFP could not be observed (S1 Fig).

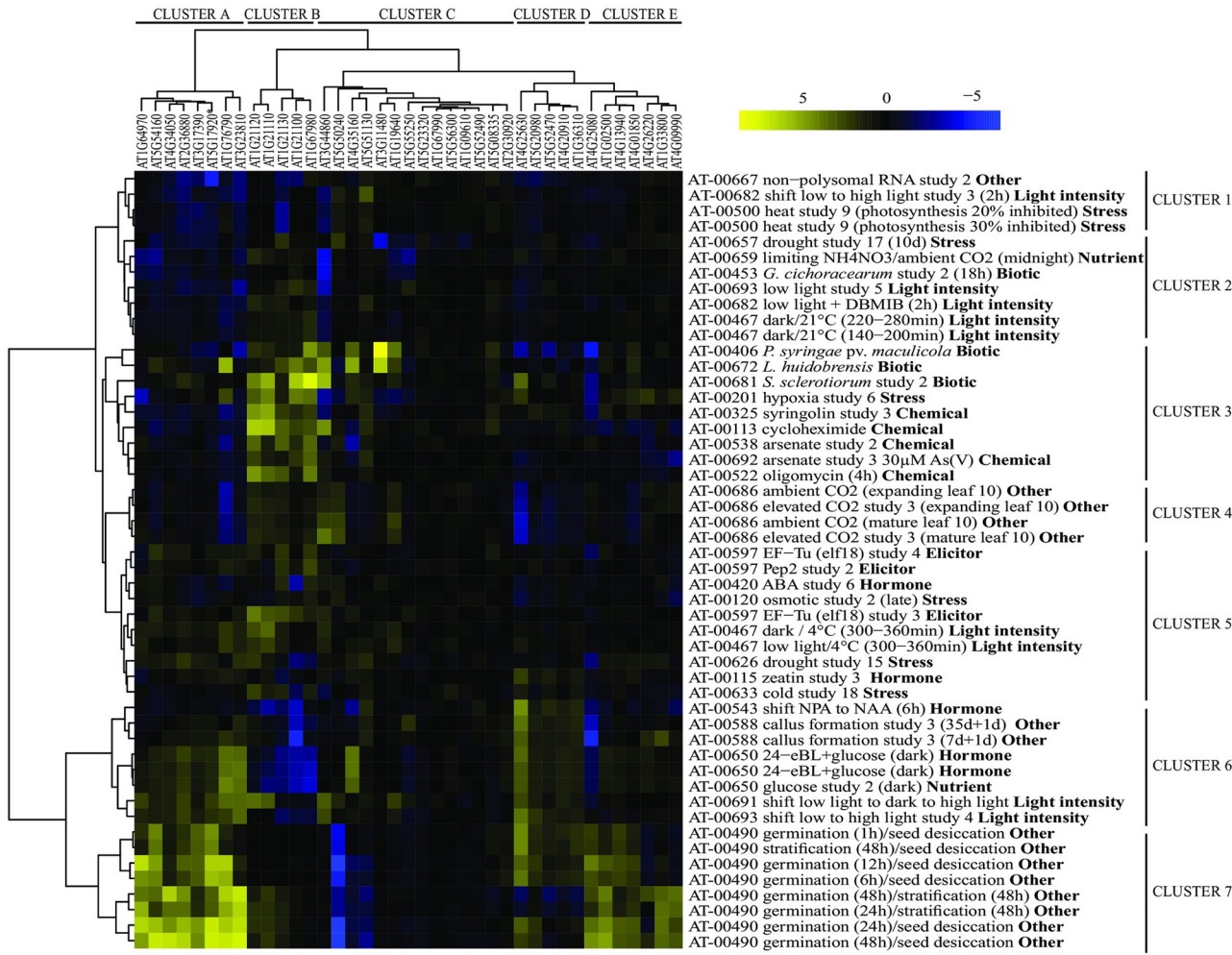

**Fig 1. Hierarchically clustered heatmap depicting dynamic adjustments in the transcript abundance for genes encoding O-methyltransferases (O-MTs) and enzymes of the activated methyl cycle in *A. thaliana*.** The analysis was performed using the "Perturbations" tool in GENEVESTIGATOR. Experiments in which at least 60% of the input genes were differently expressed were selected to build the cluster heatmap. The input genes were retrieved from UniProt reviewed database by searching for "O-methyltransferase" and filtering for species (*Arabidopsis thaliana*), and this list was combined with genes encoding the AMC enzymes (S1 Table). *This gene represents both AT5G17920 and AT3G03780 as they were indistinguishable because they share the same probe in Affymetrix Arabidopsis ATH1 microarray chip.

## Assessment of *A. thaliana* SAHH by 2D gel electrophoresis

Previously, we detected the presence of *A. thaliana* SAHH1 and SAHH2 in oligomeric compositions, including the SAHH complex 4 [17,22,23], suggesting that the SAHH isoforms are likely to form hetero-oligomeric complexes. We also found that the abundant *A. thaliana* SAHH complex 4 co-migrated with another abundant protein spot containing e.g. CARBONIC ANHYDRASE 1 (CA1; previously identified as the chloroplastic SALICYLIC ACID-BINDING PROTEIN 1 SABP3) [29, 30], whereas other abundant co-migrating protein spots were not detected on a 2D or 3D Clear Native gel systems [17,23]. The following proteomic approach was therefore designed to decipher whether CA1 forms a component in the SAHH complex 4 (Fig 3).

The stability of the SAHH-containing complexes was first assessed by using SDS as a detergent and DTT as a reducing agent. Upon treatment of soluble leaf extracts with 1% SDS, the

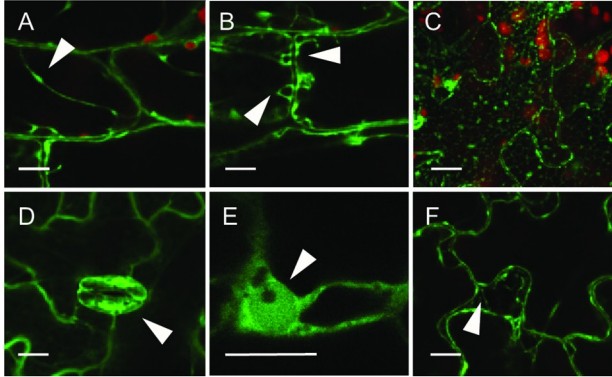

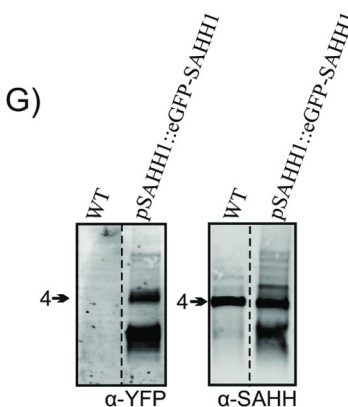

**Fig 2. Sub-cellular localization of SAHH1 in *A. thaliana* leaves.** A-F) Confocal microscopy images obtained from transgenic plants stably expressing *SAHH1p::EGFP-SAHH1*. EGFP-SAHH1 was localized to cytoplasmic strands with associated bodies (A), vesicular structures (B), reticulate constructions (C), stomatal guard cells (D), nuclei (E) and the plasma membrane (F). The red color corresponds to chlorophyll autofluorescence from chloroplasts. The scale bars correspond to10 μm. G) Immunoblot analysis depicting the presence of EGFP-SAHH1 in oligomeric protein complexes. Leaf extracts from *A. thaliana* Col-0 and a transgenic line expressing *SAHH1p::EGFP-SAHH1* were isolated, separated on Clear Native gel electrophoresis and immunodetected with anti-SAHH antibody.

SAHH complexes 1, 2 and 3 became dispersed and also the SAHH complexes 4 and 5 became less abundant when separated by CN-PAGE (S2 Fig). In contrast, treatment of soluble leaf extracts with 10 mM DTT did not affect the stability of SAHH complex 4 (S2 Fig).

Immunoblot analysis with anti-SAHH antibody suggested that pre-treatment of soluble leaf extracts with 0.25% SDS did not alter the migration of SAHH complex 4 on CN-PAGE (Fig 3A). In line with this finding, the SAHH-containing protein spot was observed on 2D CN-PAGE in both non-treated and SDS-treated samples (Fig 3B). In contrast, the GLUTA-MINE SYNTHASE 2 (GLN2) homo-octamer, which was present in the non-treated control sample, disappeared upon pre-treatment with 0.25% SDS (Fig 3B). These findings suggested that the stability of protein complexes was differentially affected by the 0.25% SDS treatment. Indeed, when the protein complexes were separated on 2D CN-PAGE, the abundant CA1-containing protein spot, which co-migrated with SAHH in control samples, no longer co-migrated with SAHH in the SDS-treated samples (Fig 3B). This finding indicated that

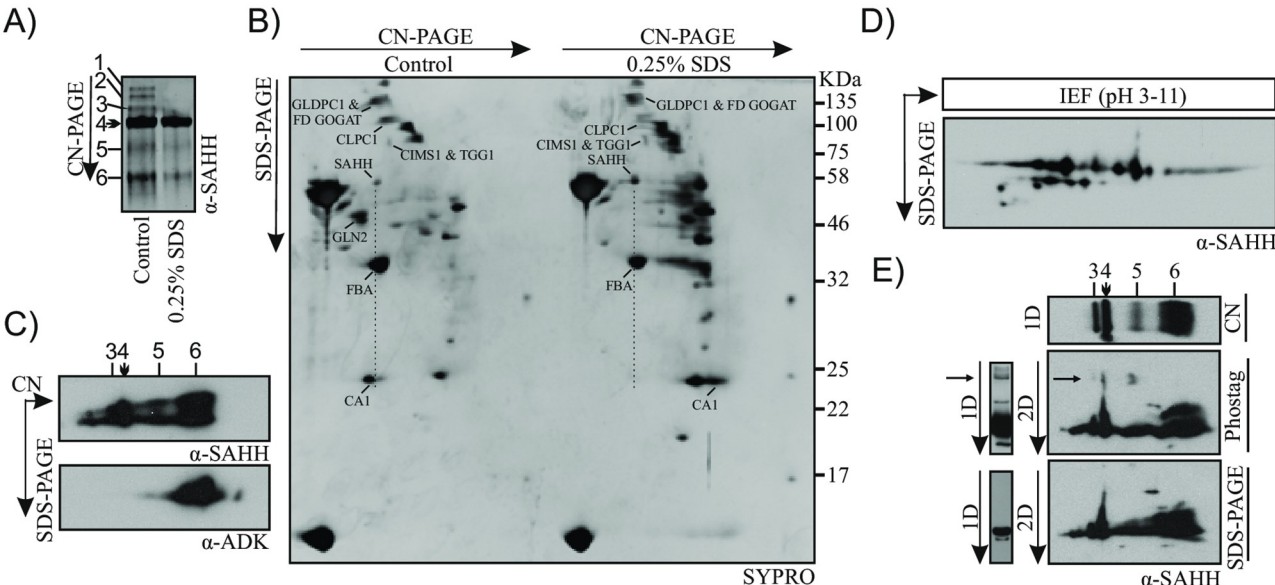

**Fig 3. Biochemical analysis of *A. thaliana* SAHH.** A) *A. thaliana* leaf extracts were incubated in the presence and absence of 0.25% SDS and thereafter separated on Clear Native (CN) gels. SAHH was detected by immunoblotting using anti-SAHH antibody. B) *A. thaliana* leaf extracts were incubated in the presence and absence of 0.25% SDS and thereafter separated on Clear Native (CN) gels followed by 12% acrylamide SDS-PAGE in the second dimension, which was stained with a total protein stain (SYPRO). The SAHH-containing major spots originating from the SAHH complex 4, as well as CARBONIC ANHYDRASE 1 (CA1), FRUCTOSE BISPHOSPHATE ALDOLASE (FBA), GLYCINE DECARBOXYLASE P-PROTEIN 1 (GLDPC1), FERREDOXIN-DEPENDENT GLUTAMATE SYNTHASE 1 (FD GOGAT), CLPC HOMOLOGUE 1 (CLPC1), COBALAMIN-INDEPENDENT METHIONINE SYNTHASE (CIMS1), THIOGLUCOSIDE GLUCOHYDROLASE 1 (TGG1), and GLUTAMINE SYNTHASE 2 (GLN2) are marked. C) *A. thaliana* leaf extracts were separated on 2D-CN PAGE and SAHH and ADK were detected by immunoblotting using anti-SAHH and anti-ADK antibodies. SAHH complex 4, which does not co-migrate with ADK, is marked with an arrow. D) *A. thaliana* leaf extracts corresponding to 100 μg of protein were fractionated by isoelectric focusing in a pH range from 3 to 11 and subsequently separated by 12% acrylamide SDS-PAGE in the second dimension. E) *A. thaliana* leaf extracts corresponding to 25 μg of protein were fractionated on CN-PAGE and the protein complexes were thereafter separated on 7.5% acrylamide Phostag-PAGE or on a similar 7.5% acrylamide SDS-PAGE lacking the Phostag reagent. SAHH complex 4 in the horizontal lane of 1D CN-PAGE is indicated by an arrowhead. SAHH immunoblots of 1D Phostag and 1D SDS-PAGE are shown in parallel to the 2D gels to indicate the SAHH band patterns on the different gel systems. The presence of a slow-migrating SAHH species on the Phostag gel is indicated by arrows.

the CA1-containing spot did not contain stoichiometric components of SAHH complex 4. The altered localization of CA1 on the 2D SDS-PAGE was likely due to monomerization of the CA1 complex by the SDS-treatment. Another abundant protein spot that co-migrated with SAHH complex 4 was identified as chloroplastic Fructose Bisphosphate Aldolase (FBA) (Fig 3B) [17], which as a chloroplastic protein does not co-localize with SAHH in the cell (Fig 2A) and is therefore highly unlikely to interact with SAHH. Moreover, immunoblot analysis of 2D CN gels did not provide evidence for co-migration of SAHH complex 4 and ADENOSINE KINASE (ADK) (Fig 3C), even though SAHH and ADK are known to interact *in planta* [18]. Based on these findings, the co-migration of SAHH1 and SAHH2 in native gel electrophoresis [22,23], and the apparent 200 kDa MW of the complex, it can be deduced that the SAHH complex 4 may be composed by a hetero-oligomeric tetramer of the enzyme, although the relative proportion of the two SAHH isoforms within these complexes remains to be established.

Next we assessed the extent to which SAHH is present in different forms in *A. thaliana* leaf extracts. Isoelectric focusing and 2D SDS-PAGE, followed by immunoblot analysis detected SAHH in multiple spots with different pIs and three different molecular masses (Fig 3D). Such variety of combinations may arise from various combinations of PTMs, which could allow enormous versatility in the regulation of SAHH function.

The final approach was designed to explore whether SAHH is differentially phosphorylated within the different oligomeric compositions. To this end, soluble leaf extracts were run on CN-PAGE, followed by separation of differentially phosphorylated proteins on Phostag gels in the second dimension (Fig 3E). In parallel, a control lane was separated on an SDS-PAGE which, similarly to the Phostag gel, was devoid of urea. Immunoblotting of the Phostag gels with anti-SAHH antibody revealed slow-migrating protein spots, indicative of SAHH phosphorylation in complexes 3, 4 and 5 (Fig 3E).

## Evolutionary conservation of SAHH in land plants

To gain insights into evolutionary conservation of SAHH, we first compared the amino acid sequences of SAHH in divergent plant species, including *A. thaliana*, *L. luteus*, *B. oleracea*, *S. oleracea* and *P. patens* (Fig 4). *A. thaliana* and *B. oleracea* are closely related species and showed 99% SAHH amino acid sequence similarity (Fig 4A and 4B, Table 1). Even between the more distantly related species, pair-wise amino acid comparison between *L. luteus* and *P. patens* SAHH indicated 90% similarity (Fig 4A and 4B, Table 1). *A. thaliana* SAHH1 has been reported to undergo phosphorylation [17, 31–33], S-nitrosylation [34,35], acetylation [31] and ubiquitination [36] at multiple sites. Majority of the experimentally described PTMs sites were

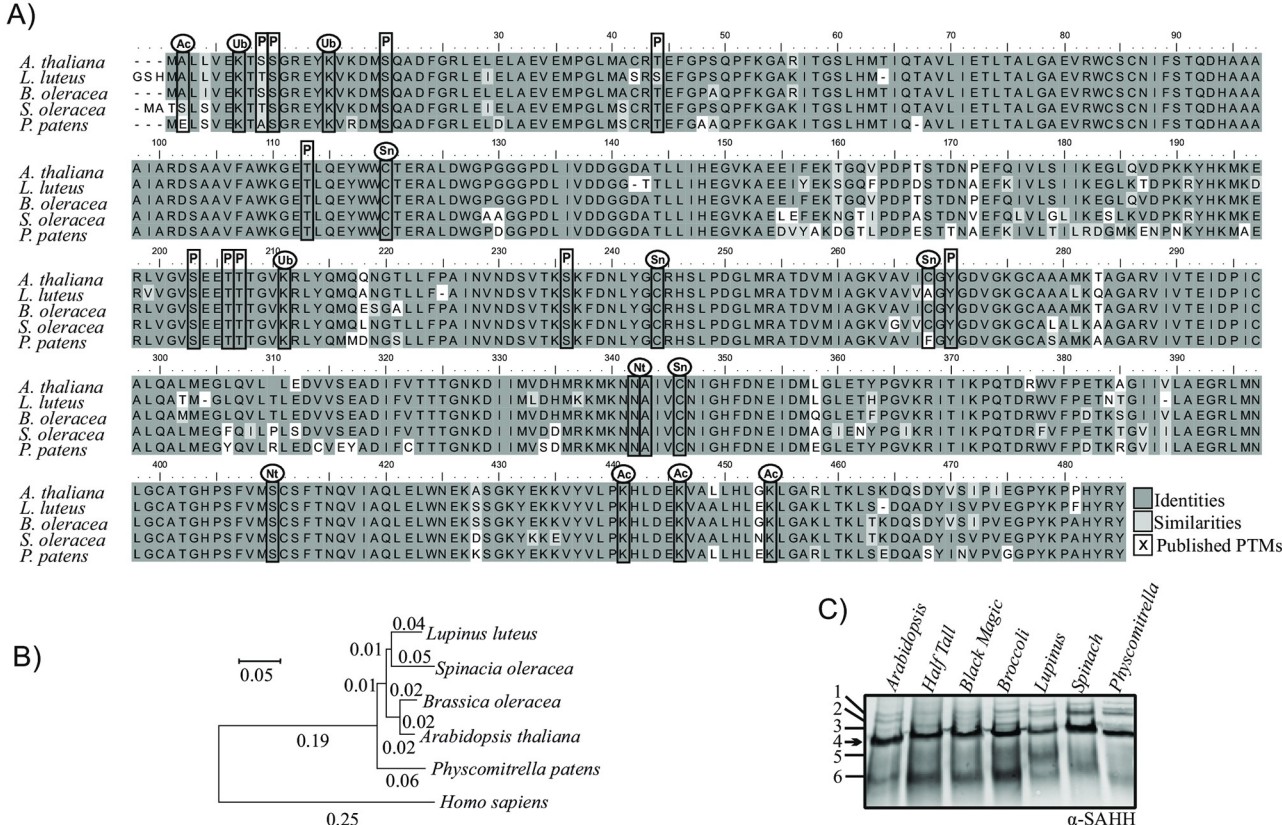

**Fig 4. Evolutionary conservation of SAHH on protein level.** A) Amino acid sequence alignment of SAHH1 between *A. thaliana*, *L. luteus*, *B. oleracea*, *S. oleracea* and *P. patens*. Ac, Lysine Acetylation; P, phosphorylation; Nt, N-terminus Proteolysis; Sn, S-nitrosylation; Ub, ubiquitination. B) Phylogenetic tree based on SAHH1 amino acid sequence constructed using the neighbor-joining method. Numbers represent substitution per amino acid based on data from 500 trees. C) SAHH containing protein complexes in evolutionarily divergent plant species as detected by anti-SAHH antibody. Total soluble protein extracts of *A. thaliana*, *B. oleracea* convar *italic* (broccoli), *B. oleracea* convar *acephala* var. Half Tall (kale) and *B. oleracea* convar *acephala* Black Magic (kale), *L. luteus*, *S. oleracea* and *P. patens* were separated on CN-PAGE. *A. thaliana* protein complexes typically detected by anti-SAHH antibody are indicated by numbers. The abundant protein complex detected by anti-SAHH antibody is indicated by arrow.

**Table 1. Pair-wise comparison of SAHH1 amino acid sequences from *A. thaliana*, *L. luteus*, *B. oleracea*, *S. oleracea* and *P. patens*. Identities and similarities are shown.**

|  | IDENTITIES | SIMILARITIES |
|---|---|---|
| *A. haliana* & *L. luteus* | 0,90 | 0,95 |
| *A. thaliana* & *B. oleracea* | **0,97** | **0,99** |
| *A. thaliana* & *S. oleracea* | 0,89 | 0,92 |
| *A. thaliana* & *P. patens* | 0,87 | 0,93 |
| *L. luteus* & *B. oleracea* | 0,90 | 0,95 |
| *L. luteus* & *S. oleracea* | 0,88 | 0,92 |
| *L. luteus* & *P. patens* | **0,85** | **0,90** |
| *B. oleracea* & *S. oleracea* | 0,89 | 0,94 |
| *B. oleracea* & *P. patens* | 0,87 | 0,94 |
| *S. oleracea* & *P. patens* | 0,85 | 0,92 |

conserved in the plant species studied (Fig 4A), suggesting that post-translational regulation of SAHH could be a conserved feature among land plants (Fig 4A).

Assessment of potential conservation of SAHH complex formation across evolutionarily distant plants by CN-PAGE and immunoblotting with the anti-SAHH antibody detected a predominant protein complex, which displayed similar apparent molecular weight as the *A. thaliana* SAHH complex 4 in all the plant species studied (Fig 4C).

To examine the abundant SAHH protein complex in spinach (*S. oleracea*) leaves, their soluble foliar proteins were separated on 2D CN-gels and the Sypro-stained map of protein spots was compared with those obtained from mass spectrometry analysis of *A. thaliana* leaves in this study (S3 Fig) and in previous reports [17,23]. Immunoblot analysis detected SAHH in a predominant protein spot that displayed similar localization in the 2D CN gel in both species (S3 Fig). The immunoblot was then overlaid on the Sypro-stained protein map, and the protein complexes that coincided with the SAHH immuno-response were excised from the 2D gels and subjected for analysis by mass spectrometry (S3 Fig). This approach identified SAHH1 and SAHH2 isoforms from the spot that originated from *A. thaliana* (S3 Fig, S3 Table). Distinct SAHH isoforms, annotated as Adenosylhomocysteinase, were also identified from the protein spot that originated from spinach (S3 Fig, S3 Table). These findings suggested that an approximately 200 kDa SAHH-containing protein complex is present also in spinach leaves (Fig 4, S3 Fig, S3 Table). Likewise, a co-migrating protein complex was also detected in 1D CN-PAGE of protein extracts isolated from *L. luteus* (Fig 4C), for which the resolved crystal structure suggested that SAHH would be active as a dimer [14,15]. Taken together, these results suggested that SAHH is a conserved enzyme, which may form similar oligomeric protein complexes in phylogenetically different land plants.

## Light-induced adjustments of SAHH in *A. thaliana* and *P. patens*

To assess stress-induced adjustments in SAHH, we exposed the two well-established, phylogenetically different model plants, *A. thaliana* and *P. patens*, to high irradiance levels for two days. Immunoblot analysis of protein complexes separated on CN-PAGE suggested a subtle but statistically significant decrease in the abundance of SAHH complex 4 in high-light-exposed *A. thaliana*. *P. patens* in turn showed a slight accumulation of the complex that co-migrated with *A. thaliana* SAHH complex 4 (Fig 5A and 5B and S4 Fig). A detectable, however statistically insignificant, decrease was also observed in the abundance of a complex denoted SAHH complex 2 in high-light-exposed *A. thaliana* (Fig 5A and 5B and S4 Fig, [17]). *P. patens*, in contrast, responded by somewhat variable, statistically insignificant increases in the

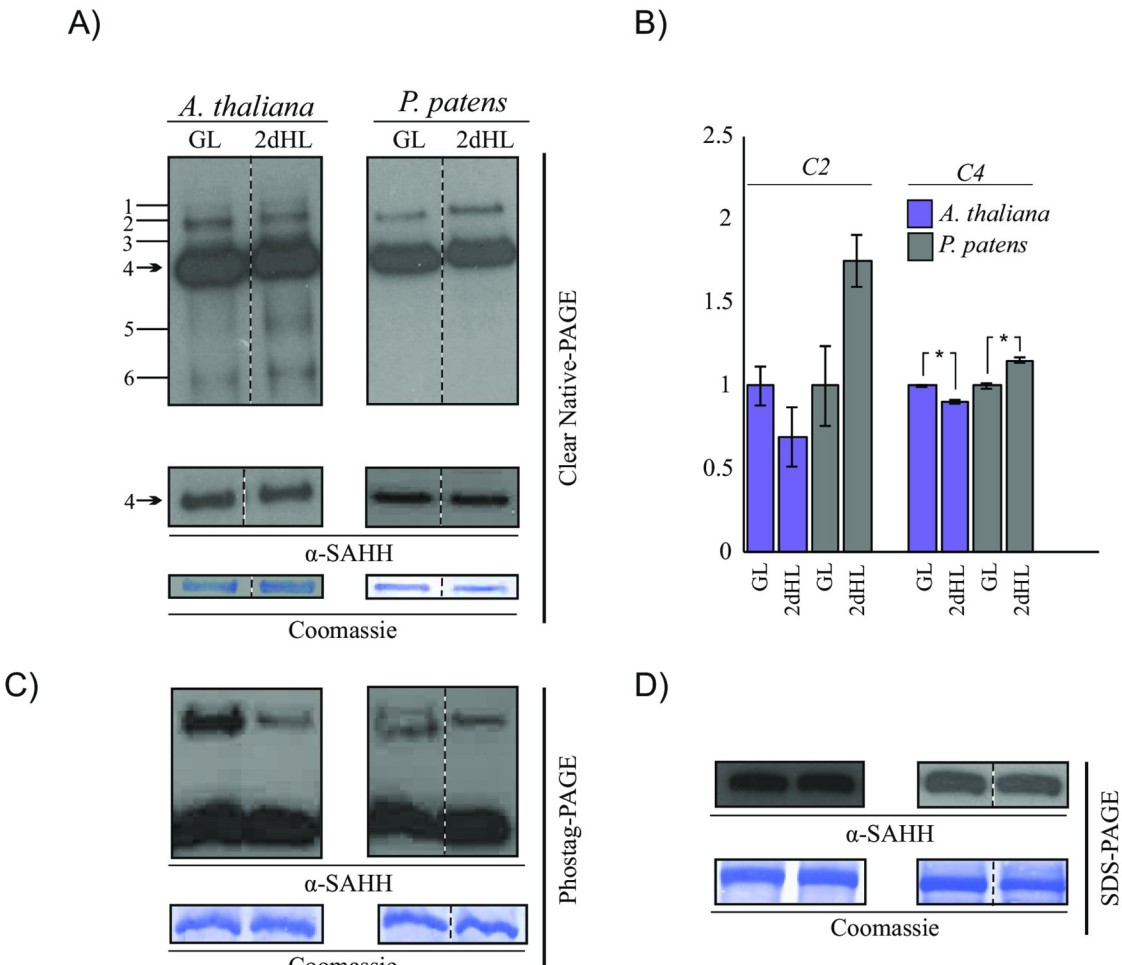

**Fig 5. Light-stress-induced adjustments in SAHH.** *A. thaliana* was grown under 130 µmol photons m$^{-2}$ s$^{-1}$ for 16 days and thereafter shifted 800 µmol photons m$^{-2}$ s$^{-1}$ for 2 days. *P. patens* as a shade-adapted moss species was grown under 45 µmol photons m$^{-2}$ sec$^{-1}$ for 13 days and thereafter illuminated under 500 µmol photons m$^{-2}$ s$^{-1}$ for two days. SAHH was separated by gel-based systems and immunodetected by using an anti-SAHH antibody. Coomassie-stained membranes are shown as loading controls for each experiment. A) Oligomeric protein complexes as detected by anti-SAHH antibody and clear native (CN)-PAGE in *A. thaliana* and *P. patens* in growth light (GL) and after 2-day illumination under high light (2dHL). Six oligomeric protein complexes detected by the anti-SAHH antibody in *A. thaliana* are indicated by numbers. The lower panel depicts an immunoblot with a shorter exposure time for visualization and quantification of the abundant SAHH complex 4. B) Quantification of protein complexes recognized by α-SAHH antibody in *Arabidopsis thaliana* and *Physcomitrella patens* in growth light and after two-day exposure to high light. C2 and C4 refer to Arabidopsis SAHH complexes 2 and 4, and complexes with similar apparent molecular weights in *P. patens*. Relative values are presented. The t-test p-values obtained were 0.053 for *A. thaliana* C2, 0,083 for *P. patens* complex that co-migrated with *A. thaliana* C2, 0.0008 for *A. thaliana* C4 and 0.001 for *P. patens* complex that co-migrated with *A. thaliana* C4. The asterisks indicate statistically significant difference at P<0.005; n = 3. C) SAHH protein phosphorylation as detected by anti-SAHH antibody and Phostag-PAGE in *A. thaliana* and *P. patens* in growth light (GL) and after 2-day illumination under high light (2dHL). D) SAHH protein abundance as detected by anti-SAHH antibody and SDS-PAGE in *A. thaliana* and *P. patens* in growth light (GL) and after 2-day illumination under high light (2dHL).

abundance of a protein complex, which immuno-reacted with the anti-SAHH antibody and co-migrated with the *A. thaliana* SAHH complex 2 in CN gels (Fig 5A and 5B, S4 Fig).

Analysis of SAHH phosphorylation by Phostag gel electrophoresis suggested light-dependent phosphoregulation, which was particularly evident in *A. thaliana* (Fig 5C, S4 Fig). In *A. thaliana*, a slow-migrating form of SAHH was detected in leaf extracts isolated from growth light conditions, while in high-light-exposed leaves such phosphorylated form of SAHH was

barely detectable (Fig 5C, S4 Fig). *P. patens* also displayed slow-migrating forms that could be detected with the anti-SAHH antibody, but changes in the intensity of the phosphorylated SAHH species were less obvious as compared to those observed in *A. thaliana* (Fig 5C, S4 Fig). The total abundance of SAHH did not differ between the treatments (Fig 5D, S4 Fig). Hence, both *A. thaliana* and *P. patens* displayed potential regulatory adjustments in SAHH, but the responses differed between the angiosperm and bryophyte models.

## Discussion

### SAHH is required to maintain trans-methylation reactions in different cellular compartments

Trans-methylation reactions are intrinsic to cellular metabolism and a prerequisite for normal plant growth and development. Reflecting the enormous diversity of species-specific trans-methylation reactions that take place in metabolic and regulatory networks, adjustments in SAHH function can be expected to differ under different physiological states in different species. SAM-dependent trans-methylation reactions occur in a multitude of subcellular compartments where SAH must be efficiently metabolized to maintain MT activities [4]. Functional characterization of *A. thaliana* mutants has demonstrated that loss of SAHH function can result in global inhibition of MT activities because of accumulation of SAH [37]. Changes in the sub-cellular localization of SAHH could also significantly affect the efficiency of specific trans-methylation reactions [18,38,39]. Associated with this, SAHH can translocate from the cytoplasm into the nucleus [18] and is also dynamically distributed in other sub-cellular compartments, including vesicular structures, reticulate constructions and the plasma membrane, but not chloroplasts or mitochondria (Fig 2).

One of the key functions of SAHH is to maintain appropriate patterns of DNA and histone methylation in the nucleus [6,38]. The nuclear localization of SAHH was recently attributed to a 41-amino-acid segment (Gly150-Lys190), which is a prerequisite for nuclear targeting of *A. thaliana* SAHH1 [18]. Intriguingly, the surface-exposed segment does not act as an autonomous nuclear localization signal *per se*, but may rather serve as an interaction domain for associations with other proteins that can direct SAHH1 into the nucleus. In line with this idea, it was proposed that physical interactions between SAHH and MTs could provide a means for targeting SAHH to appropriate subcellular compartments in order to ensure uninterrupted trans-methylation [18].

While SAHH1 has not been localized into the chloroplast (Fig 2) [18], we detected SAHH1 as a ring in the immediate vicinity around the photosynthetic organelles (Fig 2), presumably to facilitate efficient removal of SAH upon export by SAM transporters that exchange SAH for SAM synthesized in the cytoplasm [40,41]. We also found that SAHH1 can dynamically move along cytosolic strands (S1 Video), but the possible mechanisms underlying such sub-cellular movements remain to be established. Besides protein interactions with MTs, physical contact with components of the cytoskeleton or enzymatic protein complexes may direct SAHH1 to appropriate sub-cellular localizations.

SAHH isoforms can form oligomeric complexes and undergo dynamic interactions with various endogenous and exogenous proteins [2,42,43]. SAHH has been shown to physically interact with various methyltransferases, including indole glucosinolate methyltransferases [17], mRNA cap methyl-transferase [18], and caffeoyl CoA methyltransferase (CCoAOMT) [39], presumably to ensure efficient trans-methylation reactions at accurate sub-cellular sites. These interactions are likely largely determined by the availability of MTs, which appear to be transcriptionally highly responsive to endogenous and exogenous cues (Fig 1). Yang et al. [39] proposed that *in vivo* interactions between *A. thaliana* CCoAOMT7, SAHH1/SAHH2, and

SAMS form a complex for SAM synthesis to enhance the formation of ferulate in the cell wall. Interactions between SAHH isoforms and other proteins may also be affected by different combinations of post-translational modifications.

Based on crystallographic studies and studies on recombinant, non-post-translationally modified enzyme, plant SAHH was proposed to be active as a dimer [14,15]. However, our data provides evidence suggesting that in *A. thaliana* SAHH isoforms could be predominantly present in a tetramer. This was evidenced by treatment of leaf extracts with a low concentration of SDS, which did not affect the presence of SAHH complex 4 on CN gels (Fig 3A), but abolished co-migration of potential complex-forming proteins when assessed by 2D SDS-PAGE (Fig 3B). The subunit composition and physiological significance of the abundant protein complex detected by anti-SAHH antibody in various land plants, including the moss *P. patens* (Fig 4C), remains to be established. Likewise, the subunit composition of the other complexes detected by the anti-SAHH antibody remains to be uncovered. Moreover, besides formation of biochemically rather stable oligomeric complexes, SAHH is likely to undergo transient interactions that cannot be trapped by biochemical separation of protein complexes.

## SAHH is an evolutionary conserved enzyme governed by multilevel post-translational control

The metabolic centrality of SAHH is reflected by its evolutionary conservation and the high number of PTMs, including phosphorylation, S-nitrosylation, acetylation and ubiquitination, which have been experimentally verified to occur on *A. thaliana* SAHH isoforms [17,31–36]. Conservation of the PTM sites between *P. patens*, a brypohyte, and angiosperms (Fig 4A) points to multilevel post-translational regulation that can facilitate delicate metabolic responses to environmental cues. The up-stream regulatory enzymes, such as the protein kinases and protein phosphatases, N-acetyl transferases and ubiquitin ligases, however, remain almost completely unidentified. Hints to phosphoregulation of SAHH were provided by Trotta et al. and Rahikainen et al. [17,22], who provided evidence that a protein phosphatase 2A regulatory subunit PP2A-B′γ controls SAHH complex formation and the associated trans-methylation capacity of leaf cells. Two of the phosphorylated residues on *A. thaliana* SAHH1, S203 and S236, reside on conserved amino acids in the active center of the enzyme [15], and phosphorylation at these sites could therefore affect the activation state of the enzyme. The phosphorylated residues S20 and T44 of SAHH1 in turn reside on the surface of the enzyme [15], and changes in phosphorylation of these sites could impact its protein interactions and/or subcellular localization.

Whether and how the different functional aspects of SAHH respond to environmental signals in different plant species is a key question to be resolved to understand metabolic regulation in plants. Here, we explored how the biochemical characteristics of SAHH become adjusted in response to light stress, which is an important environmental factor that poses a risk of metabolic imbalance and triggers protective responses to avoid photo-oxidative damage [44–49]. Light-stress-induced metabolic adjustments beyond photosynthetic carbon metabolism have so-far remained poorly understood. Recently, Zhao et al. [50] demonstrated that a conserved signaling mechanism, where a chloroplast retrograde signal interacts with hormonal signaling to drive stomatal closure, is operational in angiosperms, mosses and ferns. Our findings suggest that high-light-induced signals may be reflected by regulatory adjustments in SAHH at the level of complex formation and phosphorylation in *P. patens* and *A. thaliana* (Fig 5).

Taken together, cellular trans-methylation reactions are largely determined by the abundance of substrate-specific MTs that require the activated methyl cycle to retain their

activity. Among AMC enzymes, the functionality of SAHH may be controlled at the level of sub-cellular localization, complex formation and post-translational modifications, which can modulate the activity and interactions between SAHH and other proteins. Jointly, these regulatory actions determine which MTs get to interact with SAHH, thereby maintaining their activity. Hence, SAHH can be considered a key determinant of trans-methylation reactions in living cells.

## Supporting information

**S1 Table. List of reviewed *Arabidopsis thaliana* O-methyltransferases retrieved from Uni-Prot and AMC enzymes used as input for GENESTIGATOR analysis.** Clusters according to the performed hierarchical cluster analysis, protein accession, entry and protein name are indicated. Activated methyl cycle enzymes (AMC) are marked in blue.
(XLSX)

**S2 Table. Settings used in confocal microscopy analysis.**
(XLSX)

**S3 Table. Lists of proteins identified from the main SAHH-containing protein spot on 2D CN-PAGE of *Arabidopsis thaliana* and *Spinacia oleracea* leaf extracts.**
(XLSX)

**S1 Fig. Control experiments for sub-cellular localization of *Arabidopsis thaliana* SAHH1.** A) Confocal microscopy image obtained from *A. thaliana* wild type plant using microscopy settings for GFP imaging. The leaf was excited at 488 nm and fluorescence was detected at 493 to 598 nm wave length. Chlorophyll fluorescence was excited at 633 nm and detected at 647 to 721 nm wave length. The red color indicates chlorophyll autofluorescence. B) Immunoblots depicting EGFP-SAHH1 in *A. thaliana* wild type (WT) and a transgenic line stably expressing *SAHH1p::EGFP-SAHH1*. Proteins were separated on SDS-PAGE, and EGFP-SAHH1 was immunodetected with an anti-YFP antibody and SAHH was detected with an anti-SAHH antibody.
(PDF)

**S2 Fig. Immunoblot depicting SAHH protein complexes after treatment of *Arabidopsis thaliana* foliar leaf extracts with SDS and/or DTT.** For combined treatments with SDS and DTT, the leaf extract was incubated in the presence of one chemical for 30 minutes, followed by addition of the other for 30 minutes.
(PDF)

**S3 Fig. 2D-approach depicting SAHH protein complexes from *Arabidopsis thaliana* wild type (WT) and *Spinacia oleracea*.** Protein complexes were separated CN-PAGE followed by 12% SDS-PAGE in the second dimension. A) Predominant protein spots as detected by immunoblot analysis using α-SAHH antibody. B) Total protein detection by SYPRO. The spots indicated as "SAHH" in *A. thaliana* and *S. oleracea* samples were excised from the gel and the presence of SAHH was confirmed by mass spectrometry as indicated in S3 Table.
(PDF)

**S4 Fig. Biological replicates for the study of light-stress-induced adjustments in SAHH presented in Fig 5.** *A. thaliana* was grown under 130 μmol photons m$^{-2}$ s$^{-1}$ for 16 days and thereafter shifted 800 μmol photons m$^{-2}$ s$^{-1}$ for 2 days. *P. patens* was grown under 45 μmol photons m$^{-2}$ sec$^{-1}$ for 13 days and thereafter illuminated under 500 μmol photons m$^{-2}$ s$^{-1}$ for two days. The gel lanes indicated by asterisks were used to construct Fig 5. A) Oligomeric

protein complexes as detected by anti-SAHH antibody and clear native (CN)-PAGE from three independent experiments. The upper panels depict immunoblots with a shorter exposure time required for visualization and quantification of the abundant SAHH complex 4. B) SAHH protein phosphorylation as detected by anti-SAHH antibody and Phostag-PAGE in *A. thaliana* and *P. patens* in growth light (GL) and after 2-day illumination under high light (2dHL). C) SAHH protein abundance as detected by anti-SAHH antibody and SDS-PAGE in *A. thaliana* and *P. patens* in growth light (GL) and after 2-day illumination under high light (2dHL).
(PDF)

**S1 Video. Dynamic movements of SAHH1p::EGFP-SAHH1 in *Arabidopsis thaliana* cells.**
(AVI)

**S2 Video. Control video composed by confocal microscopy imaging of *Arabidopsis thaliana* wild type plant using microscopy settings for GFP imaging.**
(AVI)

**S1 Raw images.**
(PDF)

## Acknowledgments

We thank Marianna Alaviuhkola for excellent assistance in microscopy and Dr. Caterina Gerotto for providing *Pyscomitrella patens* material. The confocal imaging was performed with microscopes of the Cell Imaging and Cytometry Core at the Turku Bioscience Centre, University of Turku and Åbo Akademi University. Proteomic mass spectrometry analysis were carried out at the Turku Proteomics Facility, Turku Bioscience, University of Turku and Åbo Akademi University. The facility is supported by Biocenter Finland.

## Author Contributions

**Conceptualization:** Sara Alegre, Jesús Pascual, Andrea Trotta, Saijaliisa Kangasjärvi.

**Data curation:** Sara Alegre, Jesús Pascual, Andrea Trotta.

**Formal analysis:** Sara Alegre, Andrea Trotta, Martina Angeleri, Moona Rahikainen.

**Funding acquisition:** Jesús Pascual, Saijaliisa Kangasjärvi.

**Investigation:** Sara Alegre, Jesús Pascual, Andrea Trotta, Martina Angeleri, Moona Rahikainen, Saijaliisa Kangasjärvi.

**Methodology:** Sara Alegre, Jesús Pascual, Andrea Trotta, Mikael Brosche, Barbara Moffatt.

**Project administration:** Saijaliisa Kangasjärvi.

**Resources:** Mikael Brosche, Barbara Moffatt, Saijaliisa Kangasjärvi.

**Software:** Mikael Brosche.

**Supervision:** Saijaliisa Kangasjärvi.

**Visualization:** Moona Rahikainen.

**Writing – original draft:** Sara Alegre.

**Writing – review & editing:** Jesús Pascual, Andrea Trotta, Martina Angeleri, Moona Rahikainen, Mikael Brosche, Barbara Moffatt, Saijaliisa Kangasjärvi.

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
