## [Decision Letter · Decision Letter 0]

27 Apr 2020

PONE-D-19-35012

Evolutionary conservation and multilevel post-translational control of S-adenosyl-homocysteine-Hydrolase in land plants

PLOS ONE

Dear Dr Kangasjarvi,

Thank you for submitting your manuscript to PLOS ONE. After careful consideration, we feel that it has merit but does not fully meet PLOS ONE’s publication criteria as it currently stands. Therefore, we invite you to submit a revised version of the manuscript that addresses the points raised during the review process.

please find attached reviews from three different reviewers.

With kind regards

We would appreciate receiving your revised manuscript by Jun 11 2020 11:59PM. To enhance the reproducibility of your results, we recommend that if applicable you deposit your laboratory protocols in protocols.io, where a protocol can be assigned its own identifier (DOI) such that it can be cited independently in the future. For instructions see: http://journals.plos.org/plosone/s/submission-guidelines#loc-laboratory-protocols

We look forward to receiving your revised manuscript.

Kind regards,

Evangelia V. Avramidou, PhD

Academic Editor

PLOS ONE

Journal Requirements:

'This work was financially supported by Academy of Finland project 307719 to SK, 325122 to JP, and the Academy of Finland Center of Excellence in Primary Producers 2014-2019 (307335). SA and MR were funded by the University of Turku Doctoral Programme in Molecular Life Sciences, the Turku University Foundation and the Finnish Cultural Foundation Varsinais-Suomi Regional Fund. MB was funded by the University of Helsinki.'

'This work was financially supported by Academy of Finland (www.aka.fi)  project 307719 to SK, 325122 to JP, and the Academy of Finland Center of Excellence in Primary Producers 2014-2019 (307335). SA and MR were funded by the University of Turku Doctoral Programme in Molecular Life Sciences (https://www.utu.fi/en/research/utugs/doctoral-programme-in-molecular-life-sciences). MR was also funded by the Turku University Foundation (https://www.yliopistosaatio.fi/en/) and the Finnish Cultural Foundation Varsinais-Suomi Regional Fund (https://skr.fi/en/regional-funds/varsinais-suomi-regional-fund). The funders had no role in study design, data collection and analysis, decision to publish, or preparation of the manuscript.'

Reviewers' comments:

Reviewer's Responses to Questions

**Comments to the Author**

1. Is the manuscript technically sound, and do the data support the conclusions?

Reviewer #1: Partly

Reviewer #2: No

Reviewer #3: Yes

2. Has the statistical analysis been performed appropriately and rigorously? 

Reviewer #1: N/A

Reviewer #2: No

Reviewer #3: Yes

3. Have the authors made all data underlying the findings in their manuscript fully available?

Reviewer #1: Yes

Reviewer #2: No

Reviewer #3: Yes

4. Is the manuscript presented in an intelligible fashion and written in standard English?

Reviewer #1: Yes

Reviewer #2: Yes

Reviewer #3: Yes

5. Review Comments to the Author

Reviewer #1: In this paper, the Authors describe evolutionary conservation, as well as biochemical characterization of S-adenosyl-L-homocysteine hydrolase in land plants including multi-level post-translational control of the enzyme. They found, that an oligomeric SAHH complex 4 corresponds to a homotetrameric form of SAHH. The manuscript is generally well written and the drawn conclusions are mostly supported by the experimental results. However, I have some concern about the results and conclusions related to the oligomeric state of SAHH and a composition of the complex 4. Taken together, the paper is a strong candidate for this journal, if the authors can address all of the shortcomings in the experiments and interpretations mentioned below.

MAJOR points

1. The Authors discuss on different oligomeric forms of plant SAHHs in a light of physiological (tetramer) and artificial (dimer) conditions. It should be stress in the manuscript, that the dimeric form of Lupinus luteus SAHH was established within in vivo studies (gel filtration, crystallography etc.). Also, structural studies of L. luteus enzyme were conducted with a recombinant, not post-translationally modified enzyme.

2. Did the Authors considered e.g. gel filtration analysis (or some others like Dynamic Light Scattering) to establish oligomeric state of recombinant Arabidopsis thaliana SAHH , as well as the isolated complex 4?

3. The Authors indicate a molecular mass of the complex 4 of about 200 kDa, which corresponds to the tetrameric form of plant SAHase (~210 kDa). However, this molecular mass also could correspond to the heteroligomer composed of two SAHH and two ADK molecules (see Lee et al. 2012, reference 18 in the manuscript). Did the Authors exclude a presence of ADK in complex 4?

4. Did the Authors considered SDS-PAGE separation for the isolated and purified complex 4 visualized with SYPRO or other dye? The result should be definitive for the establishment of a composition (homo- vs. heterooligomer) of complex 4 (2D gels are a bit blurred, might be affected by a presence of SDS).

5. How the authors obtained anti-SAHH antibody? Did they use recombinant SAHH or isolated complex 4 as an antigen? If the complex 4 was used, the antibody could be used in detection of other possible (if any) components of the complex.

MINOR points

1. The full and correct name of the enzyme is S-adenosyl-L-homocysteine hydrolase. The name used in the title should be replaced with the correct one.

2. Figure 3 has three panels (A-D), whereas the figure 3 caption describes panels A-C. Also the description is shifted.

Reviewer #2: This manuscript addresses an interesting issue on the regulation of SAHH in land plants, but at the moment the manuscript is premature. The experiments performed are minimal, without information on their reproducibility, and currently do not support the conclusions. Major modifications are required before publications.

Major comments:

Figure 2 and related text (methods): Please include appropriate negative control of wild-type Arabidopsis tissues imaged at the same magnification, same exposure time and brightness/contrast settings. For subcellular localization, e.g. nucleolus, please include higher magnification pictures. This should pose no problem since the authors have access to a confocal microscope. In the methods or elsewhere appropriate, please specify the exposure time and brightness/contrast settings for all pictures shown. Same comment for S1 video, show negative control using non-GFP tissues, since plant tissues can be highly auto fluorescent the wavelengths used for GFP.

Figure 3 and related text: Description of the method used for the identification of the proteins on the 2D gels cannot be found in the manuscript. Were mass spectrometric methods used? Please make sure the methods employed are described in the revision. In addition, the panels 3C and 3D are not correctly described in the text or the figure 3 legend.

Figure 4 and associated text: There are a few problems with this figure. Firstly, the identification of 6 complexes containing SAHH1 in panel C is questionable because the number of bands in each species is different. Moreover, the specificity of the �SAHH antibody has not been thoroughly investigated, and the presence of SAHH1 in these complexes has not been confirmed by any analytical methods (mass spectrometry for example), or appropriate negative controls. This is needed to support the conclusions. Without this information about the proteins actually present in the complexes, to name the most intense band seen in each lane as the same conserved “SAHH complex 4” is not an appropriate conclusion.

Figure 5 and associated text: Same as with Fig 4; without identifying these complexes, you cannot write they are the same, responding in opposite ways to light stress. You must confirm the presence of SAHH1 in each complex in each species to make the conclusion you are making, else you need to rewrite your conclusions.

Throughout the paper: There is no statistical tests performed on any data shown, and we do not know whether the data shown are representative of reproduced experiments. When investigating the changes in complex abundance, for example in Fig. 5, it would be helpful to show three experimental replicates for each species and treatment so that we have an idea of the reproducibility of the changes, and you would be able to make statistical analysis on these changes.

Minor comments:

Line 66: ….is the only known eukaryotic….

Line 142: The samples were centrifuged at 18,000 g….

Line 272-274: sentence is not understandable. Please rewrite.

Arabidopsis and Physcomitrella and other genera names should be italicized throughout the manuscript, even when the species name is omitted.

Reviewer #3: A few Comments:

1. Line 30: evolutionary conservation and multilevel post-translational control of SAHH in land plants.

-> Authors only checked phosphorylation in current manuscript. But, authors mentioned multilevel post-translational modification. Authors need to change the sentence.

2. Line 35: in the levels of protein complex formation and post-translational modification of this.

-> Authors need to change the sentence “post-translational modification” to “phosphorylation” and then trim the whole sentence.

3. Line 278: SAHH in multiple spots with different pIs and three different molecular masses (Fig 3B).

->Error found: Fig. 3B should be changed to Fig.3C.

->I do not know the positions of the complex spots. Authors need to indicate complexes 1,2,3,4,5,6 in the Figure (for example by arrows).

4. Line 285-288: In parallel, a control sample of equal protein content was separated on an SDS-PAGE devoid of urea. Immunoblotting of the Phostag gels with anti-SAHH antibody revealed slow-migrating protein spots, indicative of SAHH phosphorylation in complexes 3, 4 and 5 (Fig 3C).

->Error found: Fig. 3C should be changed to Fig.3D.

-> Again, I do not know the positions of the complex spots. Authors need to indicate non-modified complexes 1,2,3,4,5,6 in the Figure (for example by arrows). In addition, I cannot see slow-migrating protein spots. Could you please also indicate slow-migrating protein spots in the Figure or replace the Figure with new one?

5. Figure 3B makes me confuse and feel uncomfortable. Can authors explain the results more precisely but concisely?

6. Authors examined the effects of SDS and DTT on the oligimerization of SAAH. What is the meaning of these experiments in current manuscript? We can expect the effects of ionic detergents and reducing agents on protein conformation and oligomerization.

6. PLOS authors have the option to publish the peer review history of their article (what does this mean?). If published, this will include your full peer review and any attached files.

Reviewer #1: Yes: Krzysztof Brzezinski

Reviewer #2: Yes: Jean-Michel Fustin

Reviewer #3: No

---

## [Author Response · Author response to Decision Letter 0]

4 Jun 2020

Dear Editor,

Thank you for the positive response and the very helpful critical comments to our manuscript “Evolutionary conservation and multilevel post-translational control of S-adenosyl-homocysteine-Hydrolase in land plants” (PONE-D-19-35012).

In the revised version, we have now carefully addressed the reviewer’s comments and concerns, which have helped us improve the manuscript, as detailed below.

The original raw blot/gel image data are included in Supporting Information.

We would like to ask to update the financial disclosure as follows:

'This work was financially supported by Academy of Finland (www.aka.fi) project 307719 to SK, 325122 to the salary of JP, and the Academy of Finland Center of Excellence in Primary Producers 2014-2019 (307335). SA and MR received salary from the University of Turku Doctoral Programme in Molecular Life Sciences (https://www.utu.fi/en/research/utugs/doctoral-programme-in-molecular-life-sciences). MR also received salary from the Turku University Foundation (https://www.yliopistosaatio.fi/en/) and the Finnish Cultural Foundation Varsinais-Suomi Regional Fund (https://skr.fi/en/regional-funds/varsinais-suomi-regional-fund). MB was funded by the University of Helsinki (www.helsinki.fi). The funders had no role in study design, data collection and analysis, decision to publish, or preparation of the manuscript.'

We hope the new version would meet the PLOS ONE’s criteria for publication. 

Sincerely,

Saijaliisa Kangasjärvi

Responses to the reviewers:

Reviewer #1: 

In this paper, the Authors describe evolutionary conservation, as well as biochemical characterization of S-adenosyl-L-homocysteine hydrolase in land plants including multi-level post-translational control of the enzyme. They found, that an oligomeric SAHH complex 4 corresponds to a homotetrameric form of SAHH. The manuscript is generally well written and the drawn conclusions are mostly supported by the experimental results. However, I have some concern about the results and conclusions related to the oligomeric state of SAHH and a composition of the complex 4. Taken together, the paper is a strong candidate for this journal, if the authors can address all of the shortcomings in the experiments and interpretations mentioned below.

- We thank the reviewer for a positive response and the comments that help us improve the quality of the work.

MAJOR points

1. The Authors discuss on different oligomeric forms of plant SAHHs in a light of physiological (tetramer) and artificial (dimer) conditions. It should be stressed in the manuscript, that the dimeric form of Lupinus luteus SAHH was established within in vivo studies (gel filtration, crystallography etc.). Also, structural studies of L. luteus enzyme were conducted with a recombinant, not post-translationally modified enzyme.

- This is a very important point and may in fact even explain some of the experimental discrepancies. This notion has now been made both in the introduction (page 4) and in the discussion (page 23) of the revised manuscript.

2. Did the Authors considered e.g. gel filtration analysis (or some others like Dynamic Light Scattering) to establish oligomeric state of recombinant Arabidopsis thaliana SAHH, as well as the isolated complex 4?

- Unfortunately we do not have access to these methodologies in our laboratory. Hopefully future research efforts will tackle this question. 

3. The Authors indicate a molecular mass of the complex 4 of about 200 kDa, which corresponds to the tetrameric form of plant SAHase (~210 kDa). However, this molecular mass also could correspond to the heteroligomer composed of two SAHH and two ADK molecules (see Lee et al. 2012, reference 18 in the manuscript). Did the Authors exclude a presence of ADK in complex 4?

- This is a very central point and we also thought about this possibility when we first started to work on the subunit composition of SAHH complex 4. We performed various 2D and 3D clear native electrophoresis analyses combined with immunoblotting with anti-SAHH and anti-ADK antibodies. To our disappointment, it was very clear that SAHH complex 4 and ADK did not co-migrate in any of the conditions studied. We have now added this data on wild type Arabidopsis plants in the results section (page 15) and in a new Fig. 3C.

- We have also stated in the results section (page 13) that our 3D native gel electrophoresis systems (reported by Rahikainen et al 2017 Plant J) did not detect any clear co-migrating protein spots that could be part of the Arabidopsis SAHH complex 4.

- To gain more insights into the subunit composition of the SAHH complex 4, we separated spinach soluble leaf extracts on 2D clear native gels. However, similarly to the Arabidopsis proteome maps, this approach revealed a major SAHH-containing spot in spinach, whereas no co-migrating protein spots that could represent additional subunits of the SAHH complex 4 could be uncovered. This data is now included as a new Supplemental Figure 3.

4. Did the Authors considered SDS-PAGE separation for the isolated and purified complex 4 visualized with SYPRO or other dye? The result should be definitive for the establishment of a composition (homo- vs. heterooligomer) of complex 4 (2D gels are a bit blurred, might be affected by a presence of SDS).

- Unfortunately we are not able to isolate the SAHH complex 4. We have however tried various gel-based systems to get its subunit composition. The molecular weight region in which the SAHH complex 4 migrates in the CN-gel actually comprises several overlapping protein complexes and it is therefore not possible to obtain clearly resolved bands in the first dimension after protein stain. Thus, it was not possible to cut the right band from the CN-gel. 

5. How the authors obtained anti-SAHH antibody? Did they use recombinant SAHH or isolated complex 4 as an antigen? If the complex 4 was used, the antibody could be used in detection of other possible (if any) components of the complex.

- The SAHH antibody was generated against Arabidopsis SAHH1, which was produced in E. coli, so the antibody is against SAHH, not against the SAHH complex 4. Reference to the work in which the antibody was generated is now provided in the materials and methods section, page 9.

MINOR points

1. The full and correct name of the enzyme is S-adenosyl-L-homocysteine hydrolase. The name used in the title should be replaced with the correct one.

- The reviewer is right, we have corrected this mistake.

2. Figure 3 has three panels (A-D), whereas the figure 3 caption describes panels A-C. Also the description is shifted.

- The reviewer is right, we apologize for this, and have corrected this mistake in the figure legend, noticing that a new figure 3C was added.

Reviewer #2: 

This manuscript addresses an interesting issue on the regulation of SAHH in land plants, but at the moment the manuscript is premature. The experiments performed are minimal, without information on their reproducibility, and currently do not support the conclusions. Major modifications are required before publications.

- We thank the reviewer for a positive response and the critical comments that help us improve our manuscript.

Major comments:

Figure 2 and related text (methods): Please include appropriate negative control of wild-type Arabidopsis tissues imaged at the same magnification, same exposure time and brightness/contrast settings. For subcellular localization, e.g. nucleolus, please include higher magnification pictures. This should pose no problem since the authors have access to a confocal microscope. In the methods or elsewhere appropriate, please specify the exposure time and brightness/contrast settings for all pictures shown. Same comment for S1 video, show negative control using non-GFP tissues, since plant tissues can be highly auto fluorescent the wavelengths used for GFP.

- It is indeed true that Arabidopsis leaves are rich in autofluorescent compounds. However, we did not observe autofluorescence (other than chlorophyll autofluorescence from chloroplasts) when imaging wild type plants with the GFP-imaging settings used in the manuscript. Anyhow, we have now included wild type controls as supplemental figure S2A and Supplemental video S2. In addition, we show a zoomed image of the nuclei in Figure 2. The microscope settings are indicated in a new Supplemental Table 2.

Figure 3 and related text: Description of the method used for the identification of the proteins on the 2D gels cannot be found in the manuscript. Were mass spectrometric methods used? Please make sure the methods employed are described in the revision. In addition, the panels 3C and 3D are not correctly described in the text or the figure 3 legend.

- The protein spots on the 2D map were recognized based on their shape, position and intensity when compared to the 2D protein maps of Arabidopsis soluble protein extracts we have reported previously in Trotta et al [2011, Plant Physiol. 156:1464-80. doi: 10.1104/pp.111.178442], Li et al [2014, New Phytol. 202:145-60. doi: 10.1111/nph.12622.] and Rahikainen et al [2017 Plant J. 89:112-127. doi: 10.1111/tpj.13326.]. 

- We have also added description of mass spectrometry identification of proteins detected in the main SAHH-containing spot on 2D CN-PAGE. This information has now been included in the materials and methods section, page 8-9.

- Also, we apologize for the mistake in the figure legend, it has now been corrected, noticing that a new figure 3C following the suggestion by Reviewer 1 was added.

Figure 4 and associated text: There are a few problems with this figure. Firstly, the identification of 6 complexes containing SAHH1 in panel C is questionable because the number of bands in each species is different. Moreover, the specificity of the �SAHH antibody has not been thoroughly investigated, and the presence of SAHH1 in these complexes has not been confirmed by any analytical methods (mass spectrometry for example), or appropriate negative controls. This is needed to support the conclusions. Without this information about the proteins actually present in the complexes, to name the most intense band seen in each lane as the same conserved “SAHH complex 4” is not an appropriate conclusion.

- The reviewer is right, and we have now re-written this part of the results and discussion (pages 17-18 and 23) in such a way that the conclusions are supported by the experimental evidence. We now highlight a protein complex that co-migrates with Arabidopsis SAHH complex 4 and can be detected with the anti-SAHH antibody. We made this change also because it is true that even though the SAHH amino acid sequences are highly similar between the various species, we cannot be completely sure if all the detected complexes in fact represent SAHH-containing complexes. Obtaining negative controls to be run in parallel with the plant samples is not possible, since null mutation of SAHH1 is lethal (Rocha et al, 2005, Plant Cell 17(2):404-417).

- Further, to study if the abundant protein complex contains SAHH in spinach, we separated spinach soluble leaf extracts on 2D clear native gels. Similar to Arabidopsis proteome maps, this approach revealed a major SAHH-containing spot in spinach, as identified by mass spectrometry analysis. This image is now included as a new Supplemental Figure 3. 

- We would also like to note that when setting up the project, we tested whether the anti-SAHH antibody can pull down Physcomitrella SAHH. For this, we made parallel pulldowns in the presence and absence of the anti-SAHH antibody. Mass spectrometry analysis identified Physcomitrella SAHH when the reaction was performed with the anti-SAHH antibody, while the control reaction did not pull down the protein. This, together with the very high sequence similarity between Arabidopsis and Physcomitrella SAHH proteins made us conclude that we can use the antibody to assess Physcomitrella SAHH using the antibody. However, we only have one biological replicate of the pull-down assay and would therefore propose to leave this data out from the manuscript.

Figure 5 and associated text: Same as with Fig 4; without identifying these complexes, you cannot write they are the same, responding in opposite ways to light stress. You must confirm the presence of SAHH1 in each complex in each species to make the conclusion you are making, else you need to rewrite your conclusions.

- The reviewer is right, and as stated above, we have now re-written this part of the results and discussion in such a way that the conclusions are supported by the experimental evidence.

Throughout the paper: There is no statistical tests performed on any data shown, and we do not know whether the data shown are representative of reproduced experiments. When investigating the changes in complex abundance, for example in Fig. 5, it would be helpful to show three experimental replicates for each species and treatment so that we have an idea of the reproducibility of the changes, and you would be able to make statistical analysis on these changes.

- We have now indicated on page 9 that all experiments were repeated at least three times with independent biological materials. We have also quantified the protein blots and made statistical analysis of the data presented for SAHH complex abundance in Fig. 5. This is now presented in a new Supplemental figure 4. 

Minor comments:

Line 66: ….is the only known eukaryotic….

Line 142: The samples were centrifuged at 18,000 g….

Line 272-274: sentence is not understandable. Please rewrite.

Arabidopsis and Physcomitrella and other genera names should be italicized throughout the manuscript, even when the species name is omitted. 

Thank you for noticing these; they have now been corrected.

Reviewer #3: 

A few Comments:

1. Line 30: evolutionary conservation and multilevel post-translational control of SAHH in land plants.

-> Authors only checked phosphorylation in current manuscript. But, authors mentioned multilevel post-translational modification. Authors need to change the sentence.

- The word “multilevel” was removed from the abstract.

2. Line 35: in the levels of protein complex formation and post-translational modification of this.

-> Authors need to change the sentence “post-translational modification” to “phosphorylation” and then trim the whole sentence.

- The correction was made, as advised.

3. Line 278: SAHH in multiple spots with different pIs and three different molecular masses (Fig 3B).

->Error found: Fig. 3B should be changed to Fig.3C.

- Thank you for noticing this error, which has now been corrected in the revised manuscript.

->I do not know the positions of the complex spots. Authors need to indicate complexes 1,2,3,4,5,6 in the Figure (for example by arrows).

- In the isoelectric focusing approach the individual proteins are separated based on their pI in the first dimension, so there are no SAHH complexes in this figure. The different spots arise as a consequence of different pIs of the differentially post-translationally modified SAHH isoforms in the first dimension, and different MW in the second dimension SDS-gel.

4. Line 285-288: In parallel, a control sample of equal protein content was separated on an SDS-PAGE devoid of urea. Immunoblotting of the Phostag gels with anti-SAHH antibody revealed slow-migrating protein spots, indicative of SAHH phosphorylation in complexes 3, 4 and 5 (Fig 3C).

->Error found: Fig. 3C should be changed to Fig.3D.

-> Again, I do not know the positions of the complex spots. Authors need to indicate non-modified complexes 1,2,3,4,5,6 in the Figure (for example by arrows). In addition, I cannot see slow-migrating protein spots. Could you please also indicate slow-migrating protein spots in the Figure or replace the Figure with new one?

- The SAHH protein complexes are now marked on the top of the CN gel, and the slow-migrating SAHH species on the phostag gel are indicated with an arrow. 

5. Figure 3B makes me confuse and feel uncomfortable. Can authors explain the results more precisely but concisely?

- We tried our best to improve the description of this experiment in the results section, pages 15-16. Also, the meaning of this experiment is detailed below.

6. Authors examined the effects of SDS and DTT on the oligimerization of SAAH. What is the meaning of these experiments in current manuscript? We can expect the effects of ionic detergents and reducing agents on protein conformation and oligomerization.

- The meaning of these experiments was to assess if we can find an experimental condition, where proteins that normally co-migrate with SAHH on CN gels become abolished, while the SAHH complex 4 is not affected. The aim is to show that the proteins that can be abolished are not part of the more stable SAHH oligomer.

- For example, in the manuscript Figure 3B we show that by adding SDS in the sample, Carbonic anhydrase 1 no longer co-migrates with SAHH complex 4. This indicates that the Carbonic anhydrase is not a component of SAHH complex 4.

---

## [Decision Letter · Decision Letter 1]

17 Jun 2020

PONE-D-19-35012R1

Evolutionary conservation and post-translational control of S-adenosyl-L-homocysteine hydrolase in land plants

PLOS ONE

Dear Dr. Kangasjarvi,

Thank you for submitting your manuscript to PLOS ONE. After careful consideration, we feel that it has merit but does not fully meet PLOS ONE’s publication criteria as it currently stands. Therefore, we invite you to submit a revised version of the manuscript that addresses the points raised during the review process.

We look forward to receiving your revised manuscript.

Kind regards,

Evangelia V. Avramidou, PhD

Academic Editor

PLOS ONE

Additional Editor Comments (if provided):

Dear authors,

although your manuscript has been improved, the comments which are raised from one reviewer was not answered, and he proposed a second major revision round. I also agree with his comments, so please answer his comments in order to further improve your manuscript.

With kind regards

Reviewers' comments:

Reviewer's Responses to Questions

**Comments to the Author**

1. If the authors have adequately addressed your comments raised in a previous round of review and you feel that this manuscript is now acceptable for publication, you may indicate that here to bypass the “Comments to the Author” section, enter your conflict of interest statement in the “Confidential to Editor” section, and submit your "Accept" recommendation.

Reviewer #1: All comments have been addressed

Reviewer #2: (No Response)

Reviewer #3: All comments have been addressed

2. Is the manuscript technically sound, and do the data support the conclusions?

Reviewer #1: (No Response)

Reviewer #2: No

Reviewer #3: Yes

3. Has the statistical analysis been performed appropriately and rigorously? 

Reviewer #1: (No Response)

Reviewer #2: No

Reviewer #3: I Don't Know

4. Have the authors made all data underlying the findings in their manuscript fully available?

Reviewer #1: (No Response)

Reviewer #2: Yes

Reviewer #3: Yes

5. Is the manuscript presented in an intelligible fashion and written in standard English?

Reviewer #1: (No Response)

Reviewer #2: No

Reviewer #3: Yes

6. Review Comments to the Author

Reviewer #1: The Authors have addressed my comments. Therefore, the present manuscript can be accepted for publication. I have identified a few of remaining concerns/minor corrections that that also need to be made in a final revision:

1. Page 4, line 93: is “in vivo”, should be: “in vitro”.

2. Page 18, Table 1: It is sufficient to report identity/similarity values with two - three significant digit after the period.

Reviewer #2: The authors seemed to have ignored most of the comments of the reviewers. We have all asked for major revisions, and yet none of the main figures were modified. Moreover, some of their answers have brought up new problems. The manuscript still appears very rough and premature.

There is still only on replicate shown in Fig. 5. If the authors have really performed the experiments three times as they now specify, there should be no problem showing in sup info the other two replicates. Ideally, though, as I previously asked, to have 2 or 3 replicates lanes (from different biological samples) for each treatment would make the manuscript much more believable.

The statistical test used in Fig S4 is apparently written to be t-test. This is not appropriate as changes in C2 and C4 for one species are not independent (same lane) and should be analysed together.

There is no significance shown for C2 in S4 (C2 label is missing, by the way) despite more pronounced differences in bar heights, does it mean it was not significant or the significance label was also forgotten? The text does not really make it clear either. Which changes were significant? Quantification of gel lanes from Fig. 5 and statistical analyses shown in Fig. S4 are actually very important and belong to the main figure, especially since light-dependent changes in SAHH complex are mentioned in the abstract.

The authors now seem to have used MS to identify spots shown in S3, but S3 just shows a picture of a 2D gel without any annotations on which spots were analysed to confirm the presence of SAHH. The authors should be precise about what was analysed (which 2D gel, which spot(s)) this time by MS.

While the discussion sometimes goes well beyond what is actually shown in the manuscript, there is not discussion about the contribution of SAHH1 and SAHH2 in the protein complexes the authors describe. Does their antibody also recognise SAHH2? If not, why not? In mammals Ahcyl1/Ahcyl2 interacts with Ahcy and so presumably also in plants. The authors indeed mention SAHH2 in the introduction, but then completely forget about it in the results and the discussion.

If the authors really have used MS, they should have been able to differentiate between SAHH1 and SAHH2. Why not?

Really, I can't recommend this manuscript for publication at the moment.

Reviewer #3: Most of reviewers' comments were properly corrected and revised according to reviewers' points and suggestions. But I still feel few parts of the manuscript were not clear because of data quality. So I would like to tell authors one thing that authors need to prepare your data clearer next time.

Anyhow, I recommend the paper for publication.

7. PLOS authors have the option to publish the peer review history of their article (what does this mean?). If published, this will include your full peer review and any attached files.

Reviewer #1: Yes: Krzysztof Brzezinski

Reviewer #2: Yes: Jean-Michel Fustin

Reviewer #3: Yes: Hak Soo Seo

---

## [Author Response · Author response to Decision Letter 1]

24 Jun 2020

Dear Editor and Reviewers,

Thank you very much again for the very helpful critical comments to our revised manuscript.

In the "Response to the reviewers" document we present our responses to the specific points made by each reviewer, especially the Reviewer 2.

We hope the new version would be sufficiently improved to meet the PLOS ONE’s criteria for publication. 

Sincerely,

Saijaliisa Kangasjärvi

---

## [Decision Letter · Decision Letter 2]

30 Jun 2020

Evolutionary conservation and post-translational control of S-adenosyl-L-homocysteine hydrolase in land plants

PONE-D-19-35012R2

Dear Dr. Kangasjarvi,

We’re pleased to inform you that your manuscript has been judged scientifically suitable for publication and will be formally accepted for publication once it meets all outstanding technical requirements.

Kind regards,

Evangelia V. Avramidou, PhD

Academic Editor

PLOS ONE

Additional Editor Comments (optional):

Reviewers' comments:

Reviewer's Responses to Questions

**Comments to the Author**

1. If the authors have adequately addressed your comments raised in a previous round of review and you feel that this manuscript is now acceptable for publication, you may indicate that here to bypass the “Comments to the Author” section, enter your conflict of interest statement in the “Confidential to Editor” section, and submit your "Accept" recommendation.

Reviewer #2: All comments have been addressed

2. Is the manuscript technically sound, and do the data support the conclusions?

Reviewer #2: Yes

3. Has the statistical analysis been performed appropriately and rigorously? 

Reviewer #2: Yes

4. Have the authors made all data underlying the findings in their manuscript fully available?

Reviewer #2: Yes

5. Is the manuscript presented in an intelligible fashion and written in standard English?

Reviewer #2: Yes

6. Review Comments to the Author

Reviewer #2: The authors have addressed my comments adequately. The manuscript now provide more solid evidence about the conservation and regulation of SAHH and is ready for publication. The light response is particularly interesting because it would indicate that methyl metabolism as a whole is light responsive.

7. PLOS authors have the option to publish the peer review history of their article (what does this mean?). If published, this will include your full peer review and any attached files.

Reviewer #2: **Yes: **Jean-Michel Fustin

---

## [Editor Report · Acceptance letter]

6 Jul 2020

PONE-D-19-35012R2 

Evolutionary conservation and post-translational control of S-adenosyl-L-homocysteine hydrolase in land plants 

Dear Dr. Kangasjärvi:

I'm pleased to inform you that your manuscript has been deemed suitable for publication in PLOS ONE. Congratulations! Your manuscript is now with our production department. 

Kind regards, 

on behalf of

Dr. Evangelia V. Avramidou 

Academic Editor

PLOS ONE